# Protocadherin-dependent dendritic self-avoidance regulates neural connectivity and circuit function

Dimitar Kostadinov[1,2], Joshua R Sanes[1]*

[1]Center for Brain Science, Department of Molecular and Cellular Biology, Harvard University, Cambridge, United States; [2]Program in Neuroscience, Harvard Medical School, Boston, United States

**Abstract** Dendritic and axonal arbors of many neuronal types exhibit self-avoidance, in which branches repel each other. In some cases, these neurites interact with those of neighboring neurons, a phenomenon called self/non-self discrimination. The functional roles of these processes remain unknown. In this study, we used retinal starburst amacrine cells (SACs), critical components of a direction-selective circuit, to address this issue. In SACs, both processes are mediated by the gamma-protocadherins (Pcdhgs), a family of 22 recognition molecules. We manipulated Pcdhg expression in SACs and recorded from them and their targets, direction-selective ganglion cells (DSGCs). SACs form autapses when self-avoidance is disrupted and fail to form connections with other SACs when self/non-self discrimination is perturbed. Pcdhgs are also required to prune connections between closely spaced SACs. These alterations degrade the direction selectivity of DSGCs. Thus, self-avoidance, self/non-self discrimination, and synapse elimination are essential for proper function of a circuit that computes directional motion.

*For correspondence: sanesj@mcb.harvard.edu

**Competing interests:** The authors declare that no competing interests exist.

## Introduction

The geometry of a neuron's dendritic and axonal arbors is believed to be a major determinant of the neuron's role within a circuit. In some cases, the relationship is clear: in sensory systems, for example, the size and shape of a dendritic arbor determine the size and shape of the neuron's receptive field, and the degree of branching within the arbor determines how densely the field is sampled (*Lefebvre et al., 2015*). Other cases are more conjectural, and in very few cases have experiments attempted to make a causal link between particular dendritic geometries and neuronal function. Here, we address this issue by analyzing a retinal direction-selective circuit.

The phenomena we investigate are self-avoidance and self/non-self discrimination (S/NSD). In self-avoidance, sibling dendritic branches do not contact each other. Although not all neurons exhibit self-avoidance, this phenomenon has been observed in a variety of systems including sensory neurons of leech (*Hirudo medicinalis*; in which the process was first described), moth (*Manduca sexta*), fruit fly (*Drosophila melanogaster*), worms (*Caenorhabditis elegans*), and zebrafish (*Danio rerio*) (*Nicholls and Baylor, 1968*; *Yau, 1976*; *Kramer and Kuwada, 1983*; *Kramer and Stent, 1985*; *Grueber et al., 2001, 2003*; *Liu and Halloran, 2005*; *Sagasti et al., 2005*; *Smith et al., 2012*). Dendrites of olfactory projection neurons and axons of mushroom body neurons also exhibit self-avoidance in *Drosophila* (*Wang et al., 2002a*; *Zhan et al., 2004*; *Hattori et al., 2007*). In mammals, self-avoidance has been documented in cerebellar Purkinje cells and some types of retinal horizontal, bipolar, amacrine, and ganglion cells (*Montague and Friedlander, 1991*; *Wassle et al., 2009*; *Lefebvre et al., 2012*; *Matsuoka et al., 2012*). Several cell-surface proteins have been implicated in self-avoidance, including Dscam1, Turtle, Flamingo, LAR-like receptor tyrosine phosphatase, Unc-5, Unc-6 (Netrin), and Unc-40

**eLife digest** Nerve cells (or neurons) connect to one another to form circuits that control the animal's behavior. Typically, each neuron receives signals from other cells via branch-like structures called dendrites. Each specific type of neuron has a characteristic pattern of branched dendrites, which is different from the pattern of other types of neuron. Therefore, it is reasonable to imagine that the shape of these branches can influence how the neuron works; however, this idea has rarely been tested experimentally.

Different processes are known to act together to control the pattern of the branched dendrites. For example, dendrites in some neurons avoid other dendrites from the same neuron. This phenomenon is referred to as 'self-avoidance'. In some of these cases, the same dendrites freely interact with the dendrites of neighboring neurons of the same type; this is called 'self/non-self discrimination'. It is not clear, however, how these two processes influence the activity of neural circuits.

Both self-avoidance and self/non-self discrimination rely on the expression of genes that encode so-called recognition molecules. Kostadinov and Sanes have now altered the expression of these genes in mice to see the effect that disrupting these two phenomena has on a set of neurons called 'starburst amacrine cells' that are found at the back the eye. The dendrites of starburst amacrine cells generate signals when objects move across the animal's field of vision. These dendrites then signal to other starburst amacrine cells and to so-called 'direction-selective ganglion cells', which in turn send this information to the brain for further processing. The experiments revealed that these disruptions affected the connections between the dendrites. Starburst amacrine cells that lacked self-avoidance mistakenly formed connections with themselves—as if they mistook their own dendrites for those of other starburst cells. In contrast, neurons that lacked self/non-self discrimination made the opposite mistake, and rarely formed connections with each other—as if they mistook the dendrites of other starbursts for their own. Disruptions to either phenomenon interfered with the activity of the direction-selective ganglion cells.

Following on from the work of Kostadinov and Sanes, the next challenges include uncovering how the recognition molecules help with self-avoidance and self/non-self discrimination. It will also be important to examine whether the conclusions based on one type of neurons can be generalized to others that also exhibit these two phenomena.

(DCC) in invertebrates (*Baker and Macagno, 2000*; *Gao et al., 2000*; *Matthews et al., 2007*; *Long et al., 2009*; *Smith et al., 2012*) and Dscam, DscamL1, Slit, Robo, Sema6A, PlexA4, PlexA2, and gamma-Protocadherins (Pcdhgs) in mice (*Fuerst et al., 2008*, *2009*; *Lefebvre et al., 2012*; *Matsuoka et al., 2012*; *Sun et al., 2013*; *Gibson et al., 2014*). In each case, they appear to act through contact-dependent repellent mechanisms.

In some instances, processes of neurons that exhibit self-avoidance do not avoid other neurons of the same type; rather, they overlap extensively with and sometimes even form synapses on each other. Thus, these neurons appear to discriminate between their own processes, which they repel, and those of their neighbors, with which they interact (*Figure 1A*). This puzzling observation suggests that processes of nominally identical neurons are immune to the repellent forces that act within each other's arbors, a phenomenon that has been called S/NSD (*Zipursky and Grueber, 2013*). Of the molecules that mediate self-avoidance, two have also been shown to mediate S/NSD: fly Dscam1 and mouse Pcdhgs (*Hattori et al., 2007*; *Hughes et al., 2007*; *Matthews et al., 2007*; *Soba et al., 2007*; *Lefebvre et al., 2012*). While Dscam1 and Pcdhg proteins are not structurally related, they have three properties that allow them to mediate both self-avoidance and S/NSD. First, both are transmembrane recognition molecules with remarkable extracellular diversity. Alternative splicing of the Dscam1 transcripts and alternative promoter choice (*Figure 1B*) plus isoform multimerization of Pcdhgs lead to >10,000 recognition units (*Schmucker et al., 2000*; *Tasic et al., 2002*; *Murata et al., 2004*; *Schreiner and Weiner, 2010*; *Thu et al., 2014*). Second, each Dscam1 and Pcdhg isoform binds homophilically, but does not bind appreciably to other, closely related isoforms (*Wojtowicz et al., 2004*, *2007*; *Schreiner and Weiner, 2010*; *Thu et al., 2014*). Finally, in those cases where tests have been made, each neuron in a population expresses a small randomly selected subset of isoforms

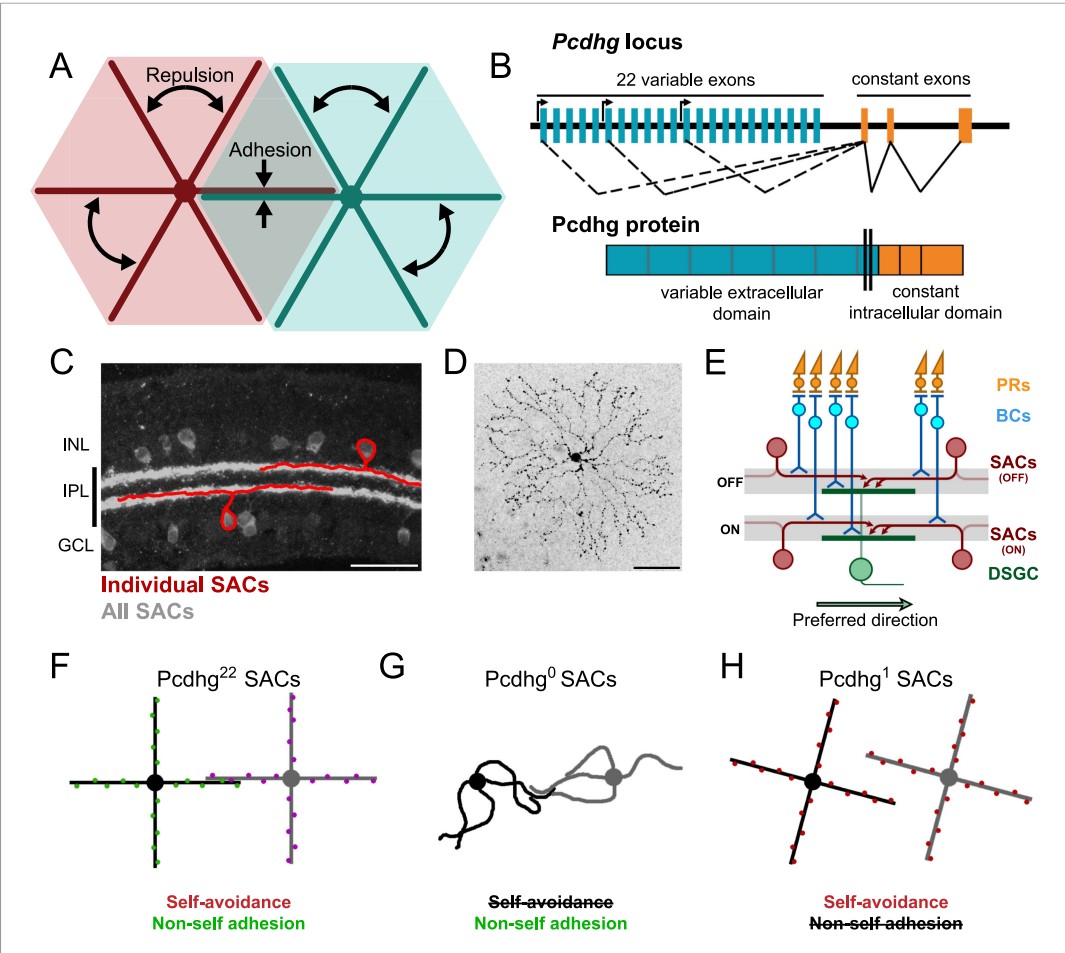

**Figure 1**. Pcdhg-dependent self-avoidance and self/non-self discrimination in SACs. (**A**) Self-avoiding neurites lack isoneuronal contacts (repulsion) but adhere to and can form synapses with neurites of other cells of the same type, displaying self/non-self discrimination (adhesion). (**B**) Schematic of *Pcdhg* genomic locus and protein product. Distinct Pcdhg isoforms are assembled by splicing one of 22 variable exons, encoding the extracellular and transmembrane portions of the protein, to three constant exons, encoding the intracellular portion of the protein. (**C**) Vertical section of retina stained against ChAT to label all SACs (gray) overlaid with cartooned individual OFF and ON SACs (red). OFF SAC cell bodies reside in the inner nuclear layer (INL) and ON SAC cell bodies reside in the ganglion cell layer (GCL). SAC neurites reside in the inner plexiform layer. (**D**) *En face* view of individual dye-filled ON SAC in Pcdhg$^{22}$ retina. (**E**) Schematic of the retinal direction-selective circuit components and connections. PRs, photoreceptors; BCs, bipolar cells; SACs, starburst amacrine cells; DSGC, direction-selective ganglion cell. Gray stripes indicate OFF and ON direction-selective sublaminae (S2 and S4, respectively). Green and red arrows indicate directional preferences of DSGCs and SAC dendrites, respectively. (**F–H**) Schematic representation of the effects of changing Pcdhg expression in SACs (summary from *Lefebvre et al., 2012*). SACs from Pcdhg$^{22}$ retinas (**F**) are posited to express unique subsets of Pcdhgs and thus exhibit both self-avoidance and non-self adhesion. SACs from Pcdhg$^{0}$ retinas (**G**) express no Pcdhgs and thus do not exhibit self-avoidance. SACs from Pcdhg$^{1}$ retinas (**H**) all express the same Pcdhg and thus exhibit self-avoidance but not non-self adhesion. Scale bar = 50 μm in **C** and **D**.

(*Neves et al., 2004*; *Zhan et al., 2004*; *Kaneko et al., 2006*; *Miura et al., 2013*; *Toyoda et al., 2014*), leading to molecular diversification that, in the case of *Drosophila* Dscam1, has been demonstrated to be important for proper patterning of neural circuits (*Hattori et al., 2009*). Together, these observations have led to a model for self-avoidance and S/NSD in which Dscam1- and Pcdhg-mediated homophilic interactions generate signals leading to repulsion. Because all dendrites (or axons) of a single neuron display the same set of Dscam1 or Pcdhg isoforms, they exhibit self-avoidance. On the other hand, any individual neuron is unlikely to encounter a neighbor that displays the same combination of isoforms, so the neurons do not repel each other and thus display S/NSD.

These morphological and molecular analyses of self-avoidance and S/NSD have led to several hypotheses about roles they might play in the function of neurons and neuronal circuits. To our knowledge, however, none of these hypotheses has been tested experimentally. Here, we report such tests, focusing on retinal starburst amacrine cells (SACs; *Figure 1C*). These neurons have planar, radially symmetric dendritic arbors that exhibit striking self-avoidance (*Figure 1D*), but they fasciculate and form synapses with neighboring SACs (*Lee and Zhou, 2006*), and thus exhibit S/NSD. SACs also provide the principal inhibitory input to ON and ON-OFF direction-selective retinal ganglion cells (DSGCs) and are essential for their direction selectivity (*Yoshida et al., 2001*). Elegant structural and functional studies have revealed the principal elements of the underlying mechanism: individual SAC dendrites are inhibitory direction-selective subunits that wire asymmetrically to DSGCs and inhibit these ganglion cells when visual motion is presented along their proximo–distal axis (*Euler et al., 2002*; *Fried et al., 2002*; *Briggman et al., 2011*; *Vaney et al., 2012*). Thus, the preferred direction of motion for the DSGC is opposite, the preferred direction of motion for the SAC dendrites that innervate it (*Figure 1E*). In addition, SACs form inhibitory synapses onto each other, and it has been suggested that these connections sharpen the directional preference of SAC dendrites and thus the directional preference of the DSGCs that they innervate (*Lee and Zhou, 2006*; *Enciso et al., 2010*; *Taylor and Smith, 2012*).

We showed recently that Pcdhgs mediate self-avoidance and S/NSD in SACs (*Lefebvre et al., 2012*). Pcdhg-deficient SACs exhibit a dramatic loss of self-avoidance but maintain overlap with neighboring SACs, as if they mistake their own dendrites for those of their neighbors and fail to repel them. In contrast, forcing all SACs to express the same single Pcdhg isoform restores self-avoidance to individual cells but decreases the overlap between neighboring cells, as if they mistake dendrites of these neighbors for their own and repel them (*Figure 1F–H*). These results lead to three specific hypotheses about circuit function: (1) in the absence of self-avoidance, SACs will form synapses with themselves (autapses), (2) when S/NSD fails, SACs will form few synapses with each other, and (3) loss of self-avoidance or S/NSD will degrade the direction selectivity of DSGCs. Here, we present evidence in support of these hypotheses, thereby providing insights into the functional roles of self-avoidance and S/NSD. We also demonstrate an unexpected role of Pcdhgs in control of synapse elimination.

## Results

### SACs are connected by inhibitory synapses

*Zheng et al. (2004)* demonstrated the presence of GABAergic synapses between SACs shortly after eye-opening in rabbits. To begin this study, we confirmed that similar connections occur in young mice and asked whether they persist in adults. In addition to releasing GABA, SACs also release acetylcholine, the only retinal neuron to do so (*Hayden et al., 1980*; *Famiglietti, 1983*), so we used a line that expresses Cre recombinase from the choline acetyltransferase locus to mark and manipulate them selectively (Chat[cre]; *Rossi et al., 2011*). We mated Chat[cre] mice to lines that express Cre-dependent fluorescent reporters (*Buffelli et al., 2003*; *Madisen et al., 2010*), identified SACs in explants, and recorded from pairs of ON SACs shortly after eye opening (postnatal day [P] 15–24; eye opening occurs at P14) and in young adults (P40-100) (*Figure 2A*). We refer to wild-type SACs as Pcdhg[22] SACs, since they have their full complement of Pcdhgs. In each case, we tested pairs separated by distances varying from 35 to 175 µm; the dendritic radius of SACs in living tissue is ~100 µm and varies little between P15 and P100 (*Figure 2—figure supplement 1A,B*). For each pair, we stepped presynaptic SACs from a holding potential ($V_h$) of −70 mV to +20 mV while holding postsynaptic SACs at +30 mV to record inhibitory currents. In the majority of cases, we obtained bidirectional recordings; we found fewer unidirectional connections between neighboring pairs than would be expected by chance (*Figure 2—figure supplement 2D*).

Stimulation of a SAC elicited an inhibitory current in a neighboring SAC in some but not all pairs tested at P15-24 and P40-100 (*Figure 2B,C*). Currents occurred with a latency of ~7 ms and averaged ~15 pA in connected cells at both ages (*Figure 2J*). They were blocked by 50 µM picrotoxin and reversed at the chloride reversal potential for our recording solutions (~−70 mV), indicating that they were GABAergic and inhibitory (*Figure 2—figure supplement 2A–C*). Although SAC–SAC connections have a cholinergic component before eye-opening in both rabbits and mice (*Zheng et al., 2004*; *Ford et al., 2012*), they exhibited no significant cholinergic component after eye opening (data not shown).

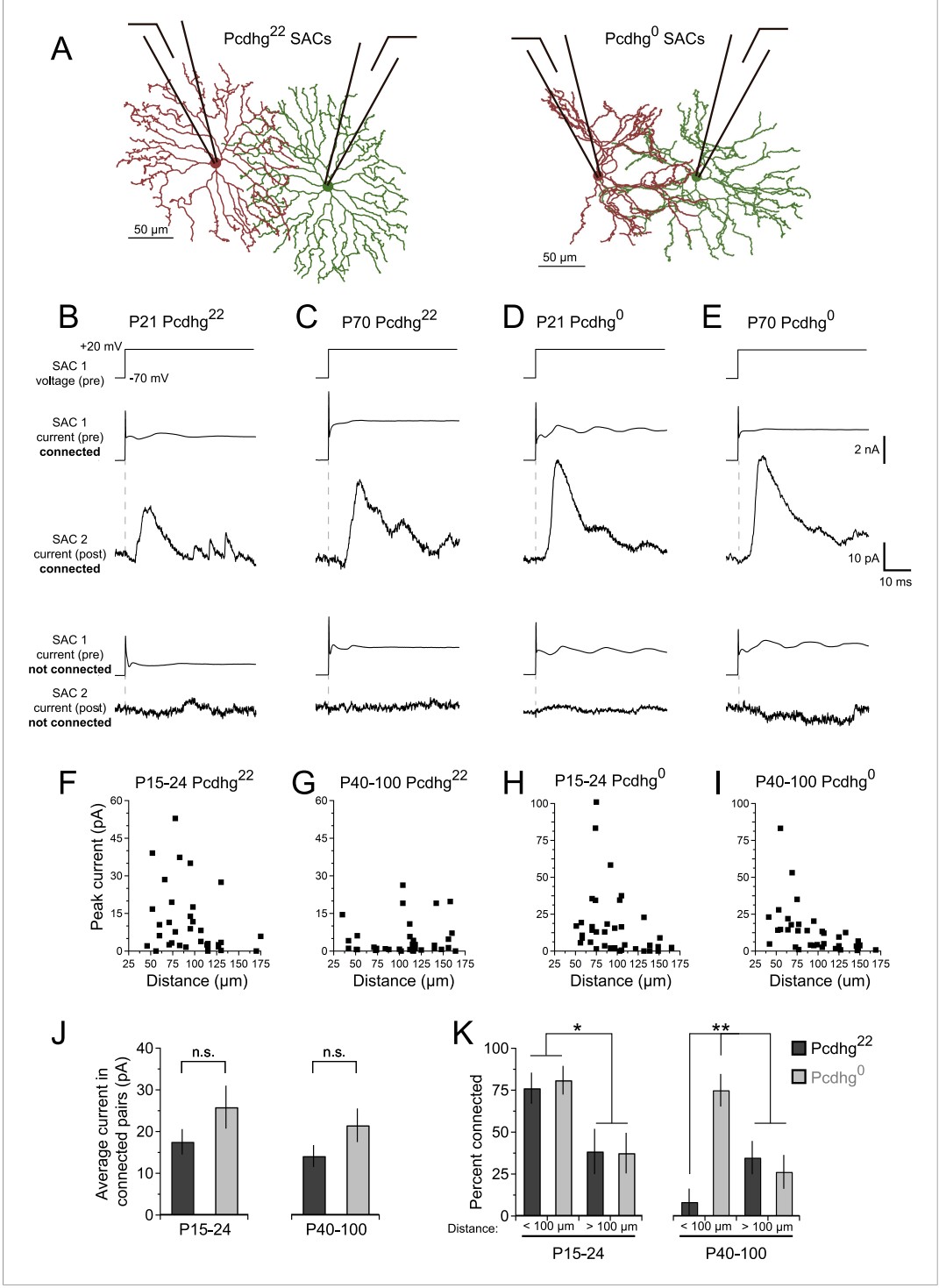

**Figure 2**. SAC–SAC connections in Pcdhg[22] and Pcdhg[0] retinas. (**A**) Paired recording configuration: SACs at various intercellular distances were targeted for recording in Pcdhg[22] (left) and Pcdhg[0] (right) retinas. Imaged are tracings of real SACs. (**B–E**) Presynaptic voltage steps from $V_h = -70$ to $+20$ mV (top) and examples of currents recorded from both pre- and postsynaptic pairs of SACs that were connected (middle) and not connected (bottom) in juvenile Pcdhg[22] retinas (**B**), adult Pcdhg[22] retinas (**C**), juvenile Pcdhg[0] retinas (**D**), and adult Pcdhg[0] retinas (**E**). (**F–I**) Scatter plots of intercellular distance vs peak current size in juvenile Pcdhg[22] retinas (**F**), adult Pcdhg[22] retinas (**G**), juvenile Pcdhg[0] retinas (**H**), and adult Pcdhg[0] retinas (**I**). Number of connections tested = 34, 35, 37, and 39 in **F–I**, respectively. (**J**) Average peak current in connected SAC pairs at P15-24 (left) and P40-100 (right). Number of
*Figure 2. continued on next page*

*Figure 2. Continued*

connections recorded = 21, 9, 23, and 20 in juvenile Pcdhg[22] retinas, adult Pcdhg[22] retinas, juvenile Pcdhg[0] retinas, and adult Pcdhg[0] retinas, respectively. (**K**) Distance-dependence of SAC–SAC connectivity at in P15-24 animals (left) and P40-100 animals (right). Data are shown as mean ± S.E.M. Statistics: n.s. = not significant, *p < 0.05, **p < 0.01. See also *Figure 2—figure supplements 1–4*.

The following figure supplements are available for figure 2:

**Figure supplement 1**. Recording distances and SAC dendritic radii.

**Figure supplement 2**. Characterization of SAC–SAC synaptic connections.

**Figure supplement 3**. Lamination and spacing of SACs are normal in Pcdhg[0] and Pcdhg[1] retinas.

**Figure supplement 4**. Normal retinal morphology in Pcdhg[0] and Pcdhg[1] retinas.

## Synapses between closely spaced SACs are eliminated after eye-opening

The frequency with which SACs were interconnected varied systematically with the distance between their somata and with age. At P15-24, pairs were over twice as likely to be connected if they were separated by 35–100 μm than if they were separated by 100–175 μm (*Figure 2F,K*, left). This difference mirrors the inverse relationship of the distance between SACs and the overlap of their dendritic arbors (*Figure 2—figure supplement 1C*). In contrast, connections were seldom detectable between pairs separated by <100 μm in adults. The frequency of connections between pairs >100 μm apart did not change significantly with age (*Figure 2G,K*, right), indicating that the decline did not reflect decreased ability to detect connections in older mice. The most parsimonious explanation for this difference is that synapses between closely spaced SACs are eliminated as SACs mature.

## Pcdhgs drive elimination of connections between closely spaced SACs

Next, we asked whether Pcdhgs are required for formation of SAC–SAC synapses. For this purpose, we inactivated all 22 Pcdhgs in SACs using a conditional Pcdhg allele (Pcdhg[flox]) (*Lefebvre et al., 2008*) and the Chat[cre] line. We refer to Pcdhg[flox/flox]; Chat[cre] mice as Pcdhg[0] and controls (Pcdhg[flox/+] or Pcdhg[+/+]; Chat[cre]) as Pcdhg[22]. Restricting mutation to SACs allowed us to analyze roles of Pcdhgs in SACs without the complication of directly affecting other synaptic partners. Moreover, deletion of Pcdhgs leads to excessive cell death in many retinal neuronal populations, but not in SACs (*Lefebvre et al., 2008*, *2012*). As expected, we observed no alterations in the density of SACs or of other retinal cells in Pcdhg[0] retinas. We further verified that the laminar position and mosaic spacing of SACs, as well as overall retinal structure, did not differ detectably between Pcdhg[22] and Pcdhg[0] retinas (*Figure 2—figure supplements 3, 4*).

At P15-24, the number and strength of SAC–SAC connections were similar in Pcdhg[22] and Pcdhg[0] retinas: in both genotypes, connections were over twice as common in closely spaced pairs than in pairs separated by >100 μm and current sizes did not differ significantly between Pcdhg[22] and Pcdhg[0] retinas (*Figure 2D,H,J*). Thus, Pcdhgs are dispensable for formation of SAC–SAC synapses. In adults, in contrast, the pattern of SAC–SAC connectivity differed between Pcdhg[22] and Pcdhg[0] mice. Synapses between closely spaced SACs were retained in mutants during the period that they were lost from controls (*Figure 2E,G,I,K*). This loss of proximal connections was selective in that the frequency and size of connections between SACs separated by >100 μm did not differ significantly between Pcdhg[22] and Pcdhg[0] mice (*Figure 2J,K*). These results reveal a requirement of Pcdhgs for synapse elimination.

## Pcdhgs prevent formation of SAC autapses

If Pcdhg[0] SAC dendrites treat other dendrites of the same SAC as if they are dendrites of other SACs, they might form autapses. To test this hypothesis, we adapted a protocol that had been used to elicit autaptic currents in cultured neurons and cortical slices (*Bekkers and Stevens, 1991*; *Bacci et al.,*

*2003*). We stimulated SACs with brief voltage steps to very positive potentials (V = +60 mV, 2–4 ms), then returned to more negative potentials (V = −20 mV) (*Figure 3A*). We confirmed that this stimulation was able to elicit synaptic release in paired recordings (*Figure 3—figure supplement 1A*). These stimuli elicited autaptic currents in ~75% of Pcdhg[0] SACs at P21-24, but in no Pcdhg[22] SACs (*Figure 3B,C*). Autaptic currents resembled SAC–SAC connections in their latencies and rise times, were blockable by application of 50 µM picrotoxin, and averaged ~20 pA in size (*Figure 3F,G* and *Figure 3—figure supplement 1B,C*). We also asked whether autapses are present in adult Pcdhg[0] SACs or whether, like synapses between closely spaced SACs in wild-type retina (see previous section), they are progressively eliminated. Autapses persisted into adulthood in Pcdhg[0] SACs with sizes and frequency similar to those observed at P21-24 (*Figure 3D–G*). Thus, one role of Pcdhg-mediated self-avoidance is to prevent formation of autapses.

## SACs that express the same Pcdhg isoform are seldom connected to each other

The proposed mechanism for Pcdhg-dependent S/NSD is that the stochastic expression of a subset of Pcdhg isoforms endows each SAC with a unique molecular identity that circumvents Pcdhg-dependent avoidance, allowing neighboring SACs to interact (*Lefebvre et al., 2012*). We hypothesized that if all SACs expressed the same Pcdhg isoform, they would treat dendrites of other SACs as if they were other dendrites of the same SAC, and form few SAC–SAC synapses. To test this idea, we used a mouse line in which a single Pcdhg isoform (PcdhgC3) can be expressed in any cell type in a Cre-dependent manner (*Lefebvre et al., 2012*). We call mice in which SACs expressed only this isoform Pcdhg[1] (Rosa-CAGS-lox-stop-lox-PcdhgC3-mCherry; Chat[cre]; Pcdhg[flox/flox]).

The overall morphology, number, and spacing of SACs, as well as overall retinal structure, were normal in Pcdhg[1] mice (*Figure 4A* and *Figure 2—figure supplements 3, 4*), and SAC dendrites formed a fine plexus within which, despite a decrease in overlap between pairs of neurons (*Lefebvre et al., 2012*), they had ample opportunity to come into close proximity to each other (*Figure 4B*). We made paired recordings from SACs in Pcdhg[1] mice at P15-24 using methods described in *Figure 2* (*Figure 4C,D*). The frequency of SAC–SAC connections in Pcdhg[1] mice was ~20% of that in Pcdhg[22] or Pcdhg[0] mice (*Figure 4F*). Similarly, current sizes in connected pairs in Pcdhg[1] mice were on average ~40% of those recorded in Pcdhg[22] or Pcdhg[0] mice (*Figure 4H*). Thus, forcing expression of the same Pcdhg isoform in all SACs decreased their connection strength to <10% (0.2 × 0.4) of controls. A similar decrease was observed in adult Pcdhg[1] retinas (*Figure 4E,G,H*). We conclude that Pcdhg diversity is required for functional connectivity between neighboring SACs.

## Pcdhgs are dispensable for connections of SACs with bipolar and ganglion cells

Having found that manipulation of Pcdhg expression affects the ability of SACs to form synapses on their own dendrites or those of other SACs, we asked whether such manipulations affect their ability to receive synapses from bipolar cells or form synapses onto DSGCs. We used visual stimuli based on previous findings that the main visually-evoked excitatory input to SACs is from bipolar cells, and that SACs provide the main inhibitory input to DSGCs (*Figure 5A*) (*Taylor and Wassle, 1995*; *Vaney et al., 2012*; *Helmstaedter et al., 2013*; *Hoggarth et al., 2015*).

To assess bipolar input to SACs, we recorded from ON SACs while holding the cells at $V_h = −70$ mV and presenting bright spot flashes centered on the soma of the recorded cell. SACs received strong excitatory inputs in Pcdhg[22], Pcdhg[0], and Pcdhg[1] mice, with no significant differences among them (*Figure 5B,C*).

Four populations of ON-OFF DSGCs have been described, each tuned to one of the cardinal directions: dorsal, ventral, nasal, and temporal (*Barlow and Hill, 1963*; *Oyster and Barlow, 1967*; *Elstrott et al., 2008*). Their physiological properties other than preferred direction are similar, but they exhibit molecular differences that allow them to be marked selectively (*Kay et al., 2011*). To assess SAC input to DSGCs, we used a transgenic line in which DSGCs that prefer motion in the ventral direction express GFP (HB9-GFP; *Trenholm et al., 2011*). We introduced this transgene into the Pcdhg[22], Pcdhg[0], and Pcdhg[1] backgrounds, and recorded inhibitory currents ($V_h = 0$ mV) from GFP-labeled DSGCs, which we call vDSGCs here. Sizes of both ON and OFF inhibitory responses to spot flashes were indistinguishable across the three genotypes (*Figure 5D,E*). Similarly, excitatory

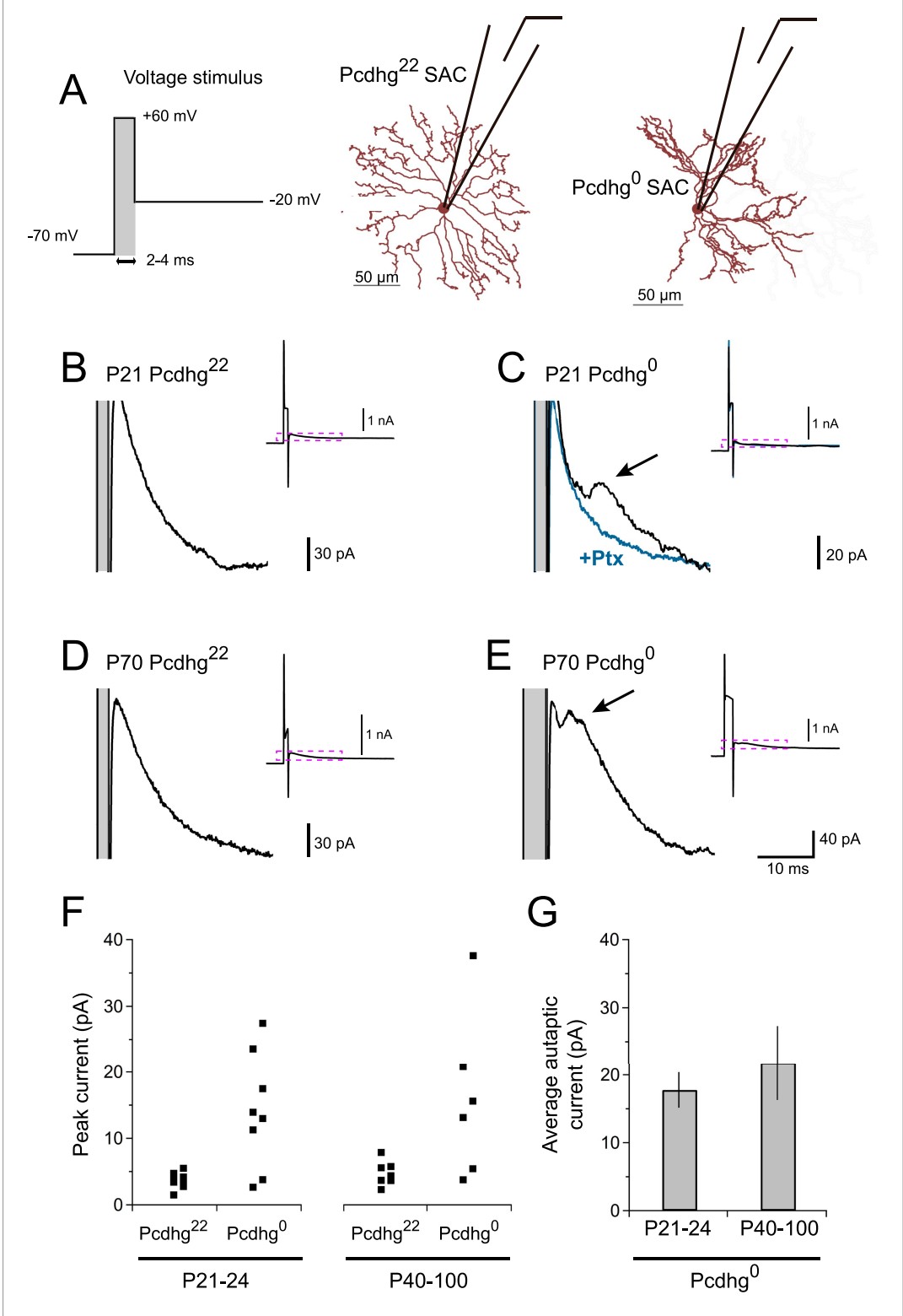

**Figure 3**. Pcdhg[0] SACs form autapses. (**A**) SAC autaptic voltage stimulus (left). Single SAC recording configuration in Pcdhg[22] (middle) and Pcdhg[0] (right) retinas. (**B**–**E**) Example currents recorded from SACs in juvenile Pcdhg[22] retinas (**B**), juvenile Pcdhg[0] retinas (**C**), adult Pcdhg[22] retinas (**D**) and adult Pcdhg[0] retinas (**E**) in response to voltage stimulus shown in **A**. Arrowheads in **C** and **E** points to autaptic currents in SAC from Pcdhg[0] retinas that were blocked by 50 μM picrotoxin (blue trace in **C**). Gray bars indicate depolarization steps to +60 mV (stimulus artifacts) that were

*Figure 3. Continued*

2 ms long in both **B**, **C**, and **D**, and 4 ms long in **E**. The shorter latency in **E** likely reflects the longer depolarization step. Full traces are shown as insets with enlarged regions outlined in magenta. (**F**) Peak outward currents measured during falling phase recorded current after initial voltage step to +60 mV. Data points are staggered slightly for visual clarity. Number of SACs recorded = 8, 8, 7, and 6 in juvenile Pcdhg[22] retinas, juvenile Pcdhg[0] retinas, adult Pcdhg[22] retinas, and adult Pcdhg[0] retinas, respectively. (**G**) Average peak autaptic currents evoked in SACs from Pcdhg[0] retinas at P21-24 (left) and P40-100 (right) at $V_h = -20$ mV. Data are shown as mean ± S.E.M. See also ***Figure 3—figure supplement 1***.

The following figure supplement is available for figure 3:

**Figure supplement 1**. Quantification of autaptic currents.

spot flash responses in vDSGCs were unaffected by manipulation of Pcdhgs in SACs (***Figure 5F,G***). Thus, Pcdhg expression in SACs is dispensable for their ability to form and maintain synapses with other cell types.

We also assessed the structure of vDSGCs in Pcdhg[22], Pcdhg[0], and Pcdhg[1] mice. We filled single cells with fluorescent dye, stained SACs with antibodies to ChAT, and imaged the two cell types. In all conditions, the ON and OFF dendrites of these DSGCs stratified in the ON and OFF SAC plexus, fasciculated with SAC dendrites, and maintained their modest structural asymmetry (***Figure 5H–J***). Thus, altering Pcdhg expression in SACs had no detectable effect on the morphology of vDSGCs. We also validated that altering Pcdhg expression in SACs did not affect cell number, mosaic spacing, or expression patterns of vDSGCs (***Figure 5—figure supplement 1***).

## Loss of SAC self-avoidance or S/NSD degrades direction selectivity of DSGCs

We next tested the hypothesis that loss of self-avoidance or S/NSD degrades the information-processing ability of SACs within the direction-selective circuit. To this end, we recorded spikes from vDSGCs while moving a bright bar over their receptive field in 8 different directions. Because vDSGCs are all tuned to a single direction in wild-type mice, we were able to ask whether manipulation of Pcdhgs affects preferred direction as well as the degree of direction selectivity.

vDSGCs in Pcdhg[22] mice exhibited strong ON and OFF directional responses (***Figure 6A***) as shown previously (***Kim et al., 2010***; ***Trenholm et al., 2011***; ***Duan et al., 2014***). We calculated a direction-selective index (DSI) for each vDSGC by computing the vector sum of the responses to different directions (***Kim et al., 2008***) and calculated both the magnitude of directional responses and the angle of preference (***Figure 6B***). Direction selectivity of vDSGCs was diminished in both Pcdhg[0] and Pcdhg[1] retinas but in different ways. In both genotypes, the average magnitude of the DSI vector was significantly decreased with respect to controls (by ~50% in Pcdhg[0] and ~35% in Pcdhg[1]; ***Figure 6C–G***). In contrast, responses of vDSGCs in Pcdhg[0] retinas exhibited a significantly greater scatter around the ventral axis than those in wild-type retinas, whereas vDSGCs in Pcdhg[1] retinas were as precisely tuned to ventral motion as controls (***Figure 6C–F,H*** and ***Figure 6—figure supplement 1***). This variance likely reflects the contorted morphology of SAC dendrites in Pcdhg[0] but not Pcdhg[1] retinas. Likewise, the variation between the preferred direction of ON and OFF responses was greater in Pcdhg[0] retinas than in either Pcdhg[22] or Pcdhg[1] retinas, indicating that SAC morphology and connectivity are disrupted independently in the ON and OFF SAC layers (***Figure 6I***).

To obtain a single measure of how well vDSGCs reported on ventral motion, we projected the directional vectors onto the ventral axis. This gave us a ventral DSI that combined the degree of directional preference and the fidelity of ventral preference for ON and OFF responses together. vDSGCs in Pcdhg[22] (control) retinas were most ventrally selective, followed by those in Pcdhg[1] retinas; vDSGCs in Pcdhg[0] retinas were the least selective (***Figure 6J***). Together, these results demonstrate that manipulating Pcdhg expression in SACs, and thereby attenuating self-avoidance or S/NSD, degrades the direction selectivity of DSGCs. Recently, ***Sun et al. (2013)*** also showed that morphological alterations of SACs disrupt directional tuning of DSGCs.

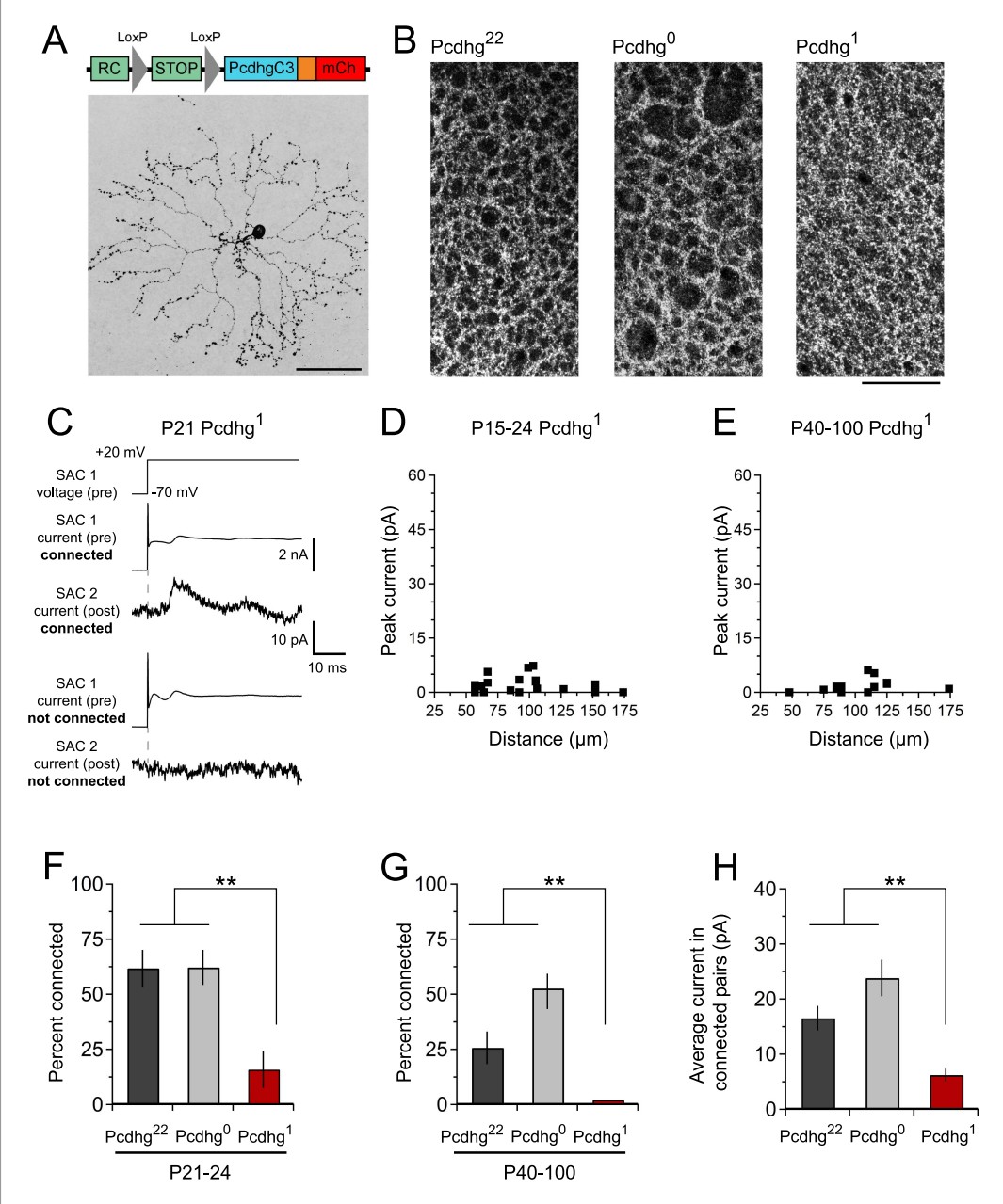

**Figure 4**. Decreased SAC–SAC connections in Pcdhg[1] retina. (**A**) Replacement of all 22 Pcdhgs in SACs with a single Pcdhg isoform (top) rescues self-avoidance in individual SACs (bottom). (**B**) Plexus of all SAC dendrites (stained with anti-ChAT) in Pcdhg[22] (left), Pcdhg[0] (middle), and Pcdhg[1] (right) retinas. (**C**) Presynaptic voltage steps from $V_h = -70$ to $+20$ mV (top) and examples of currents recorded from both pre- and postsynaptic pairs of SACs that were connected (middle) and not connected (bottom) in juvenile Pcdhg[1] retinas. (**D–E**) Scatter plots of intercellular distance vs peak current size in juvenile (**D**) and adult (**E**) Pcdhg[1] retinas. (**F**) Percent of P15-24 recorded SAC pairs that were connected, irrespective of intercellular distance. Number of connections tested = 34, 37, and 19 in Pcdhg[22], Pcdhg[0], and Pcdhg[1] retinas, respectively. (**G**) Same as **F** for adult retinas. Number of connections tested = 35, 39, and 13 in Pcdhg[22], Pcdhg[0], and Pcdhg[1] retinas, respectively. (**H**) Average peak current in connected SAC pairs at all ages. Number of recorded connections = 30, 43, and 3 in Pcdhg[22], Pcdhg[0], and Pcdhg[1] retinas, respectively. Scale bar = 50 μm in **A** and 25 μm in **B**. Data are shown as mean ± S.E.M. Statistics: **p < 0.01.

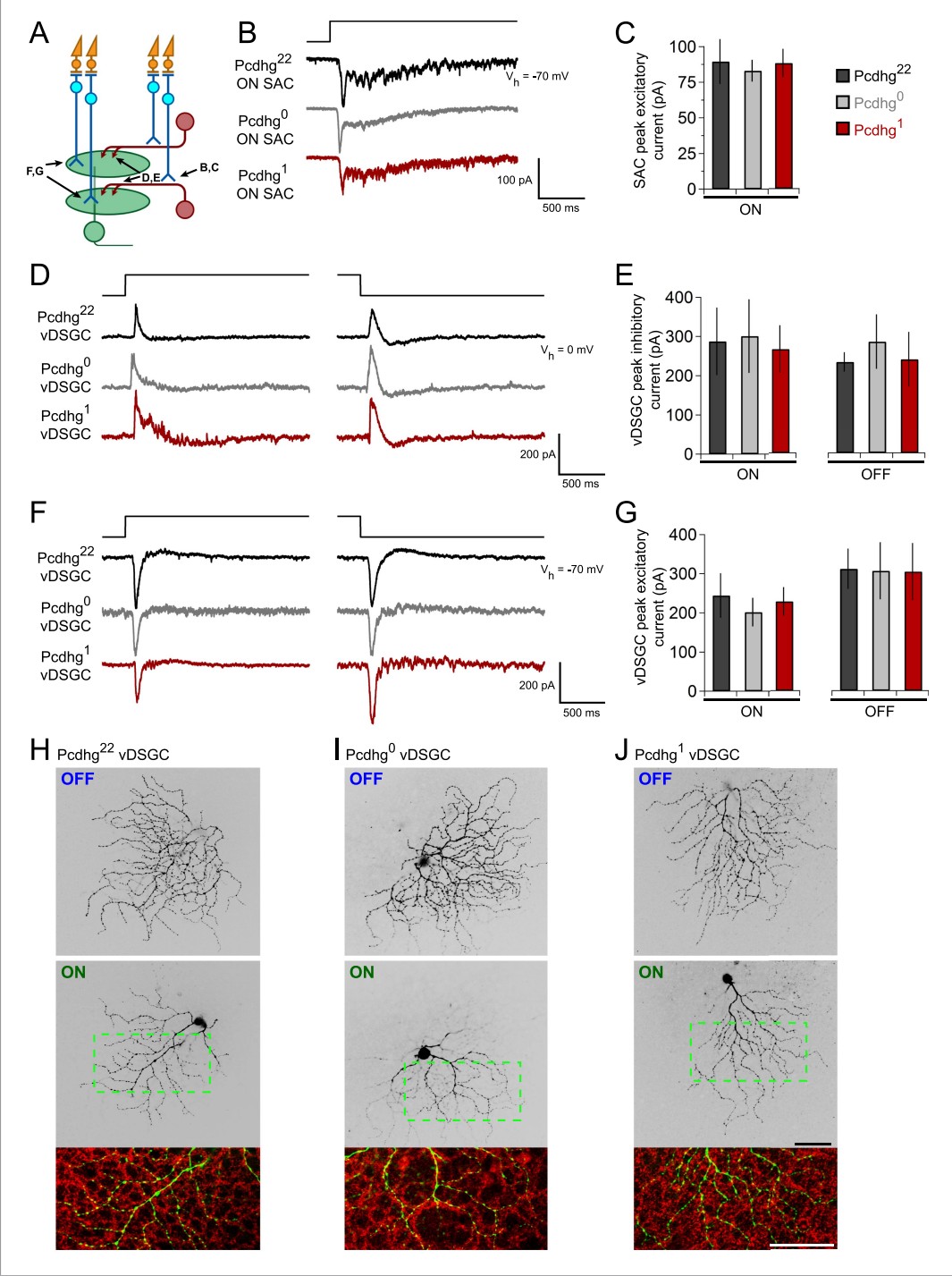

**Figure 5**. Integration of SACs into a direction-selective circuit is Pcdhg-independent. (**A**) Schematic of excitatory and inhibitory synaptic inputs of retinal direction-selective circuit, showing bipolar inputs to SACs (measured in **B** and **C**), SAC inputs to DSGCs (measured in **D** and **E**), and bipolar inputs to DSGCs (measured in **F** and **G**). (**B**) Example excitatory currents ($V_h = -70$ mV) of ON SACs from Pcdhg$^{22}$ (black), Pcdhg$^0$ (gray), and Pcdhg$^1$ (red) retinas evoked by a bright spot flash. (**C**) Average peak current responses to the onset of flash stimulus. Number of SACs recorded is 8, 9, and 7 in Pcdhg$^{22}$, Pcdhg$^0$, and Pcdhg$^1$ retinas, respectively. (**D**) Example inhibitory currents ($V_h = 0$ mV) of vDSGCs from Pcdhg$^{22}$, Pcdhg$^0$, and Pcdhg$^1$ retinas evoked by the onset (left) and offset (right) of a bright spot flash. (**E**) Average peak current responses to the onset (left) and offset (right) of flash stimulus. Number of vDSGCs recorded is 12, 13, and 10 in Pcdhg$^{22}$, Pcdhg$^0$, and Pcdhg$^1$ retinas, respectively. (**F**) Example excitatory currents ($V_h = -70$ mV) of vDSGCs from Pcdhg$^{22}$, Pcdhg$^0$, and Pcdhg$^1$ retinas evoked by the onset (left) and offset (right) of

*Figure 5. continued on next page*

*Figure 5. Continued*

a 2 s bright spot flash. (**G**) Average peak current responses to the onset (left) and offset (right) of flash stimulus. Number of vDSGCs recorded is 14, 11, and 13 in Pcdhg[22], Pcdhg[0], and Pcdhg[1] retinas, respectively. (**H–J**) Dye-filled vDSGCs with OFF and ON arborizations separated (top and middle, respectively) in Pcdhg[22] (**H**), Pcdhg[0] (**I**), and Pcdhg[1] (**J**) retinas. Bottom panels: Overlay of ON vDSGC dendrites (green) with ON SAC dendrites labeled with anti-ChAT antibody (red). Similar co-fasciculation was seen for OFF dendrites. Scale bar = 50 µm. Data are shown as mean ± S.E.M. Spot flashes were displayed for 2 s in each case. See also *Figure 5—figure supplement 1*.

The following figure supplement is available for figure 5:

**Figure supplement 1**. Normal expression, spacing, and number of vDSGCs in Pcdhg[0] and Pcdhg[1] retinas.

---

Previous studies have shown that direction-selective responses are already present at eye opening in mice but become more selective with age (*Elstrott et al., 2008*; *Yonehara et al., 2011*; *Chan and Chiao, 2013*; *Chen et al., 2014*). We wondered whether this improvement of direction selectivity with age was related to the loss of proximal SAC–SAC connections. To assess this possibility, we recorded from direction-selective responses from vDSGCs at P15-24 in Pcdhg[0] mice, which do not go through a developmental change in SAC–SAC connectivity. We confirmed the improved age-dependent directional tuning of DSGCs in control retinas. In contrast, direction selectivity of vDSGCs did not improve in Pcdhg[0] retinas (*Figure 6—figure supplement 2*). This result is consistent the idea that developmental refinement in SAC–SAC connectivity contributes to age-dependent improvement in direction selectivity.

## Synaptic mechanisms underlying effects of Pcdhgs on direction selectivity

Finally, we sought to explain the degradation of directional selectivity of vDSGCs in Pcdhg[0], and Pcdhg[1] retinas (*Figure 6*) in terms of alterations in SAC connectivity (*Figures 2–4*). To this end, we recorded inhibitory and excitatory currents from vDSGCs in the three genotypes in response to bars moving in the null and preferred directions (dorsal and ventral, respectively). As noted previously, the inhibitory currents arise predominantly from SACs, which are genetically altered in mutants, while the excitatory currents arise predominantly from bipolar cells, which are not altered.

Studies in mice and rabbits have revealed two key aspects of SAC–DSGC connectivity that lead to direction selectivity (*Fried et al., 2002*; *Taylor and Vaney, 2002*; *Vaney et al., 2012*; *Yonehara et al., 2013*; *Park et al., 2014*), both of which we confirmed in vDSGCs from Pcdhg[22] retinas. First, inhibitory input to DSGCs is greater for movement in the null direction (dorsal for vDSGCs) than for movement in the preferred direction (ventral for vDSGCs), whereas excitatory input is similar for movement in both directions (*Figure 7A,J,K*). Second, excitatory and inhibitory currents recorded from DSGCs arise at the same time when motion is in the null direction, whereas inhibitory currents lag with respect to excitatory currents when motion is in the preferred direction (*Figure 7B,L,M*). Together, these features allow inhibition to veto excitation in DSGCs more strongly for null motion than for preferred motion. Consequently, net depolarization in DSGCs is largest for motion in the preferred direction.

We found that both of these contributors to direction selectivity were blunted in Pcdhg[0] and Pcdhg[1] retinas (*Figure 7D,E,G,H* and *Figure 6—figure supplement 1*). First, inhibitory currents were larger for ventral motion and smaller for dorsal motion in Pcdhg[0] and Pcdhg[1] retinas than in Pcdhg[22] retinas, with no significant change in excitation (*Figure 7J,K*). The difference from control values was greater for Pcdhg[0] than for Pcdhg[1] retinas but significant in both. Second, the delay of inhibition in response to preferred motion was less in Pcdhg[0] and Pcdhg[1] retinas than in Pcdhg[22] retinas, with no significant change for movement in the null direction; in this case, Pcdhg[0] and Pcdhg[1] retinas were equally affected (*Figure 7L,M*). Thus, the ability of inhibition to veto excitation for preferred motion was greater in Pcdhg[0] and Pcdhg[1] retinas than in Pcdhg[22] retinas. It is likely that the differences in the size and timing of inhibitory currents in vDSGCs from Pcdhg[0] and Pcdhg[1] retinas result in the changes in spiking observed in *Figure 6*. In the Discussion, we suggest a possible explanation for these alterations in terms of perturbations in SAC self-avoidance, S/NSD, and synapse elimination.

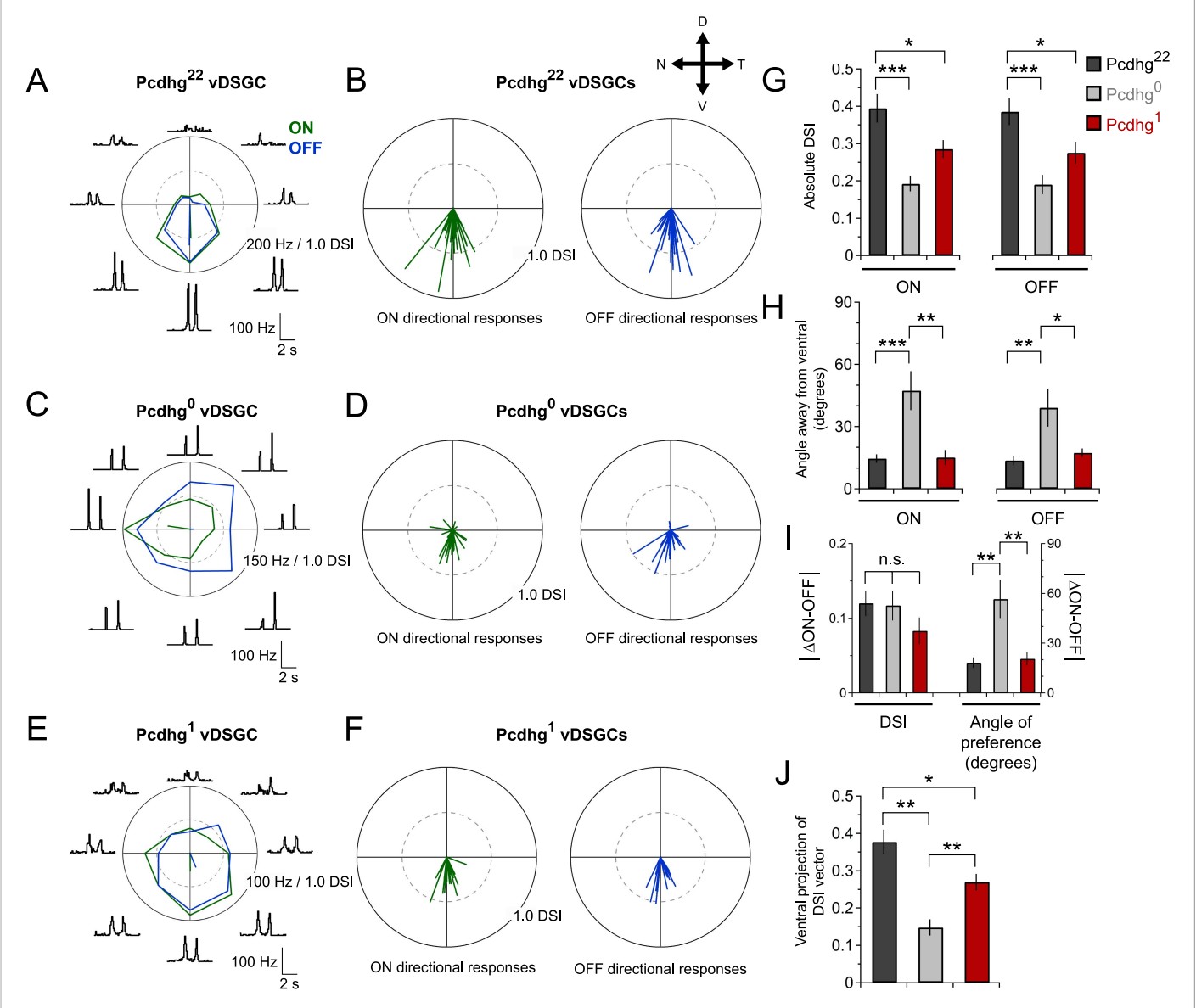

**Figure 6**. Alteration of Pcdhg expression degrades direction selectivity. (**A**) Spiking responses of vDSGC from adult Pcdhg[22] retina to a bright moving bar moving in 8 directions. Polar plot is of peak firing rates in response to bar entering (ON, green) and exiting (OFF, blue) the receptive field center. Vectors represent vector sum direction-selective indices (DSIs) of ON and OFF responses. Surrounding central plots are spike histograms used to make polar plot and calculate DSIs and preferred directions. (**B**) ON (left, green) and OFF (right, blue) DSI vectors for all recorded DSGCs in Pcdhg[22] retina (n = 28 cells). Axes of retina are indicated with compass arrows: D, V, N, and T represent dorsal, ventral, nasal, and temporal. (**C**, **D**) Same as **A** and **B** but from adult Pcdhg[0] retinas (n = 28 cells). (**E**, **F**) Same as **A** and **B** but from adult Pcdhg[1] retinas (n = 19 cells). (**G**) Mean absolute DSI for all cells recorded, irrespective of which direction they preferred. (**H**) Mean angle deviated from ventral direction for all cells recorded. (**I**) Mean absolute difference between DSI (left) and angle of preference (right) for all recorded cells. (**J**) Plot of mean ventral projections of DSI vectors. For each recorded vDSGC in **J**, maximal ON and OFF firing rates in each direction were summed and used to generate a single DSI vector for each cell. Data are shown as mean ± S.E.M. Statistics: n.s. = not significant, *p < 0.05, **p < 0.01, ***p < 0.001. See also *Figure 6—figure supplements 1, 2*.

The following figure supplements are available for figure 6:

**Figure supplement 1**. ON and OFF direction responses of vDSGCs are similarly blunted when Pcdhg expression in SACs is altered.

**Figure supplement 2**. Age-dependent improvement in direction selectivity of vDSGCs requires Pcdhgs.

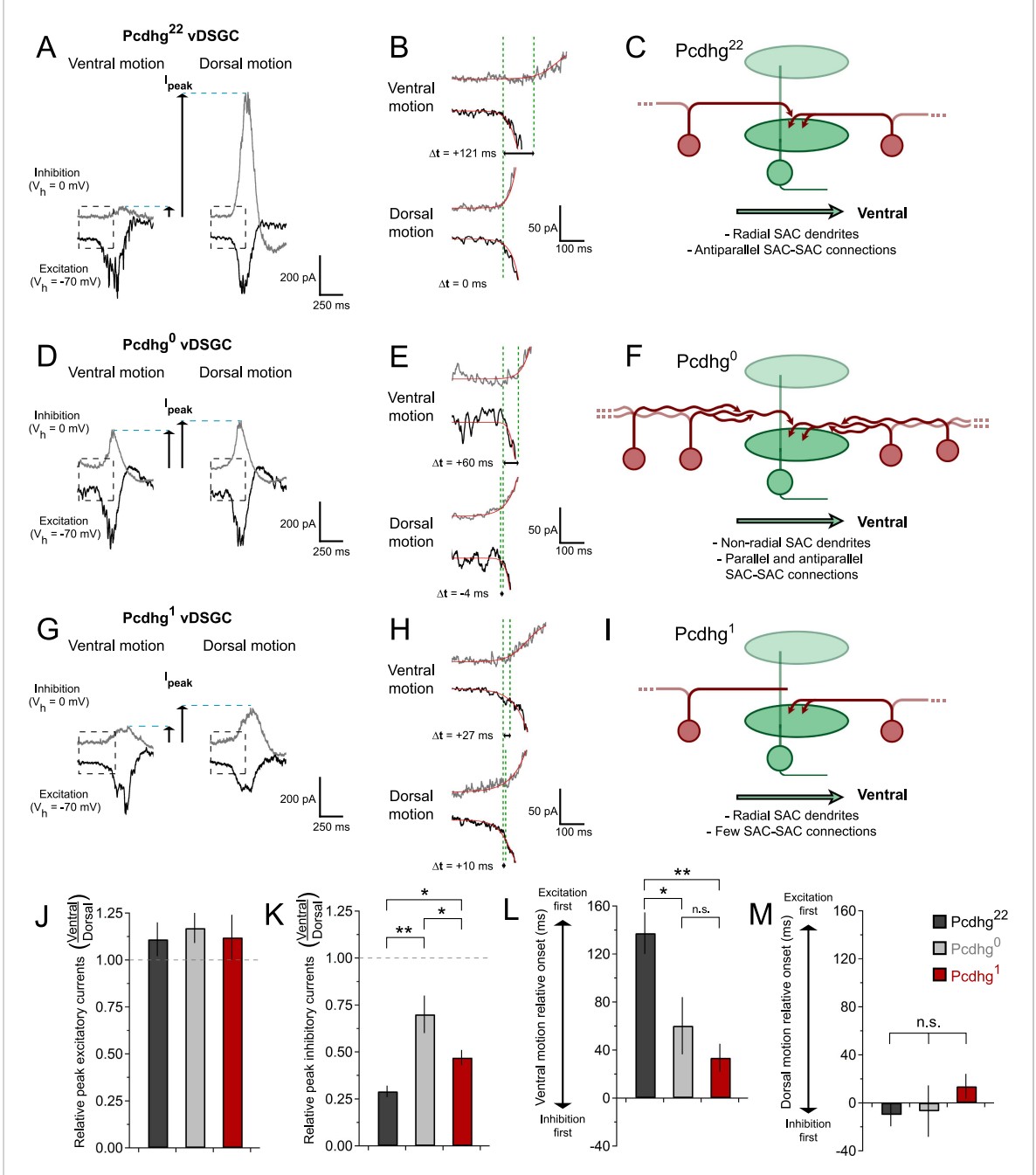

**Figure 7**. Synaptic basis of degraded direction selectivity in Pcdhg[0] and Pcdhg[1] retinas. (**A**) Example excitatory (black, $V_h = -70$ mV) and inhibitory (gray, $V_h = 0$ mV) currents evoked by leading edge (ON response) of bar moving in ventral (left) and dorsal (right) directions in vDSGC from Pcdhg[22] retina. (**B**) Examples of relative timing of excitation and inhibition in same cell from panel **A**. (**C**) Schematic of inhibitory input to vDSGCs in Pcdhg[22] retinas. vDSGCs receive inhibitory input from SAC dendrites with predominately dorsal orientations and directional preferences, setting the null direction of vDSGCs. These SAC dendrites, in turn, receive inhibitory input from SAC dendrites with predominately ventral orientation and preference, suppressing inhibition to vDSGCs during ventral motion through inhibition of inhibition. (**D**, **E**) Same as **A**, **B** but in Pcdhg[0] retina. (**F**) Schematic of inhibitory input to vDSGCs in Pcdhg[0] retinas. vDSGCs receive inhibitory input from curvilinear SAC dendrites with disrupted orientations and directional preferences, diminishing their ability to set the null direction of vDSGCs. These SAC dendrites, in turn, receive inhibitory input from both parallel and antiparallel SAC dendrites. (**G**, **H**) Same as **A**, **B** but for trailing edge (OFF response) in Pcdhg[1] retina. (**I**) Schematic of inhibitory input to vDSGCs in Pcdhg[1] retinas. vDSGCs receive inhibitory input from SAC dendrites with predominately dorsal orientations and directional preferences, setting the null direction of vDSGCs. These SAC dendrites, however, are no longer inhibited by SAC dendrites with predominately ventral orientation and preference, so their input to vDSGCs during ventral motion is not suppressed. (**J**) Ratio of peak excitatory current sizes evoked in vDSGCs by ventral vs dorsal motion in Pcdhg[22] (black), Pcdhg[0]

*Figure 7. continued on next page*

*Figure 7. Continued*

(gray), and Pcdhg[1] (red) retinas. (**K**) Same as **J** but for inhibitory currents. (**L**) Relative timing of onset of excitation compared to inhibition during ventral motion in Pcdhg[22] (black), Pcdhg[0] (gray), and Pcdhg[1] (red) retinas. (**M**) Same as **L** but during dorsal motion. Data are shown as mean ± S.E.M. Number of recorded vDSGCs = 14, 10, and 13 in Pcdhg[22], Pcdhg[0], and Pcdhg[1] retinas. Leading edge (ON) and trailing edge (OFF) responses were measured for all cells and used as independent data points for quantification. Statistics: n.s. = not significant, *p < 0.05, **p < 0.001.

## Discussion

Dendritic arbors of many neuronal types in both vertebrates and invertebrates exhibit self-avoidance and S/NSD (references in Introduction). In this study, we used SACs to assess the functional consequences of perturbing these processes. SACs were uniquely suited for this study for several reasons. First, they exhibit robust self-avoidance and S/NSD. Second, Pcdhgs are necessary for both processes, providing a means to manipulate them. Moreover, our genetic methods allowed us to manipulate Pcdhgs selectively in SACs, without directly affecting other neuronal types to which they connect. Third, removal or replacement of Pcdhgs in SACs allowed us to perturb self-avoidance and S/NSD independently. Finally, the role of SACs in retinal circuit function is remarkably well understood. By exploiting these features, we elucidated roles of self-avoidance and S/NSD in SAC connectivity, discovered a previously undescribed phase of synapse elimination between SACs, and showed that alterations in these processes decrease the ability of the retina to compute direction of motion.

### Linking Pcdhg expression to SAC connectivity

Loss of Pcdhgs has been shown to have several effects on developing neurons including decreased neuronal survival in retina and spinal cord, decreased synaptic maintenance in spinal cord, decreased dendritic branching in neocortex, and decreased self-avoidance in retina and cerebellum (*Wang et al., 2002b*; *Weiner et al., 2005*; *Lefebvre et al., 2008, 2012*; *Prasad et al., 2008*; *Garrett et al., 2012*). Any of these phenotypes would complicate our attempt to assess roles of self-avoidance and S/NSD in SAC and circuit function. We therefore manipulated Pcdhg expression selectively in SACs and performed a variety of control experiments to assess whether our manipulations affected other aspects of retinal development or function. Our results are as follows:

First, SACs are unusual among retinal neurons in that their survival does not depend on Pcdhg expression (*Lefebvre et al., 2012*), and we confirmed that SAC number was unaltered in Pcdhg[0] and Pcdhg[1] retina. Second, we confirmed (*Lefebvre et al., 2012*) that alteration of Pcdhg expression in SACs had no effect on their dendritic length or mosaic spacing. Third, basic electrical properties (resting membrane potential and input resistance) of SACs were preserved in Pcdhg[0] and Pcdhg[1] retinas (data not shown). Fourth, manipulation of Pcdhgs in SACs had no detectable effect on the strength of the inputs they receive from bipolar cells or deliver to DSGCs. Fifth, removing or replacing Pcdhgs in SACs had no detectable effect on cell number or general organization of the retina. Finally, we detected no alteration in the number, spacing, dendritic arbors, molecular markers or electrical properties of ventrally-preferring DSGCs. Thus, although we cannot completely exclude the possibility that Pcdhg manipulation had additional effects, we favor the explanation that alterations in SAC connectivity and circuit function documented here result from perturbation of Pcdhg-dependent self-avoidance, S/NSD, and synapse elimination.

### Self-avoidance, self/non-self discrimination, and SAC connectivity

Morphological studies led to the idea that self-avoidance serves to optimize coverage of a receptive field by a dendritic arbor, minimizing gaps, and clumps (*Kramer and Kuwada, 1983*; *Kramer and Stent, 1985*; *Grueber and Sagasti, 2010*). Our physiological studies revealed an additional role of self-avoidance in SACs: it prevents formation of autapses (*Figure 3*). In many neuronal types, autapses cannot form because pre- and postsynaptic machinery are confined to axons and dendrites, respectively, which are physically segregated. SACs, in contrast, form dendro-dendritic synapses, and therefore have pre- and postsynaptic specializations intermingled. This situation is not uncommon in the retina and elsewhere in the central nervous system, such as the olfactory bulb (*Murthy, 2011*). We suggest that self-avoidance may play similar roles in other such cells.

S/NSD is generally viewed as a means of limiting inter-dendritic repulsion to sibling processes, so that neurons of a single type can share territory (*Grueber and Sagasti, 2010*; *Zipursky and Grueber, 2013*; *Lefebvre et al., 2015*). In the retina, it additionally allows formation of synapses between SACs. Several types of neurons have been shown to form homotypic connections in cortex and cerebellum (*Pfeffer et al., 2013*; *Rieubland et al., 2014*). Since most molecules described to date that mediate self-avoidance are ill-suited to mediate S/NSD, additional mechanisms likely remain to be discovered. In addition, some cell types that connect homotypically may lack robust mechanisms for self-avoidance. Indeed, cortical fast-spiking interneurons, which form homotypic connections, also form autapses (*Bacci et al., 2003*). It is unclear whether these autapses are beneficial to the circuit or whether they are an acceptable cost of homotypic connectivity.

## Age- and distance-dependent elimination of SAC–SAC connections

Zhou and colleagues previously demonstrated inhibitory SAC–SAC synaptic connections in rabbit retina soon after eye opening, a result we confirmed here for mouse (*Zheng et al., 2004*; *Lee and Zhou, 2006*). We also discovered two additional features of these connections. First, in mature retina (>P40), SACs separated by less than 100 μm seldom formed synapses with each other, whereas SACs separated by > 100 μm were connected frequently. Since dendritic overlap is inversely proportional to the distance between SACs, this distance-dependence is not a passive consequence of proximity but instead implies spatial selectivity to SAC–SAC connections. Second, we found that this distance-dependence was absent in immature retinas (P15-24; eye opening occurs at P14). Thus, connections between closely spaced SACs are selectively lost as the retina matures.

We view the loss of proximal SAC–SAC connections as synapse elimination, a process that occurs in many and perhaps most neuronal types (*Kano and Hashimoto, 2009*) but has not previously been described for SACs. In most cases, synapse elimination was first described physiologically (*Redfern, 1970*; *Crepel et al., 1976*; *Purves and Lichtman, 1980*) as we have done here. For these cases, morphological confirmation was obtained many years later. We expect this will be the case for SACs as well. Such demonstration will be difficult, however, because SAC dendrites are so thin and densely packed that it is infeasible to map synapses on them by light microscopic methods. Ultrastructural studies using genetic tags or extensive reconstruction at several developmental time points will therefore be needed to decide this issue.

Why might connections between closely spaced SACs be counterproductive? Inhibitory connections between nearby SACs would frequently be made between dendrites with similar directional preferences. The ability of a SAC dendrite to respond to centrifugal motion along its dendrite would thereby decrease, because this motion would lead to inhibition of the dendrite by other SACs. This, in turn, would degrade the direction selectivity of DSGCs (*Taylor and Smith, 2012*). In contrast, connections of distant SACs will most frequently be made between dendrites with opposite directional preference; as discussed in the next section, this enhances directional computation.

Conversely, might there be a role for connections between closely spaced SACs early in development? In fact, strong SAC–SAC connectivity is critical for the developing visual system, because it underlies propagation of the retinal waves that pattern the segregation of binocular input in retinorecipient areas such as the superior colliculus and lateral geniculate nucleus (*Ford et al., 2012*; *Ackman and Crair, 2014*; *Burbridge et al., 2014*). Because waves occur before eye-opening, directional selectivity is unimportant. Thus, we suggest that postponing distance-dependent elimination of SAC–SAC connections until after eye-opening allows both the dense connectivity needed for wave propagation and the selective anti-parallel connectivity needed for direction selectivity. Consistent with this view, the direction selectivity of DSGCs increases during the period in which connections between closely spaced SACs are being eliminated.

We also found that connections between closely spaced SACs are not eliminated in the absence of Pcdhgs, revealing a novel role for these molecules in neural development. The mechanism of this effect remains to be determined. One attractive possibility is that an uneven distribution of Pcdhgs within SACs might confine synapses to distal portions of dendrites.

## Roles of SAC–SAC inhibition in directional computation

We have argued that alterations in SAC connectivity in Pcdhg$^0$ and Pcdhg$^1$ retinas documented in the first part of this study (*Figures 2–4*) result from defects in self-avoidance, S/NSD, and synapse

elimination. We now argue that these defects largely explain the degradation in direction selectivity in vDSGCs documented in the second part (*Figures 6, 7*).

As described above, SACs contribute to the direction selectivity of DSCGs in two ways. First, inhibitory currents are larger during null motion than preferred motion. The difference in inhibitory currents arises in large part from the geometric arrangement of SAC–DSGC connections: vDGSCs, for example, receive most SAC input from dendrites that respond preferentially to dorsal (null) motion (*Briggman et al., 2011*). In addition, anti-parallel inhibitory connections between SACs decrease the currents that these dendrites would otherwise provide during preferred motion (*Figure 7C*). Together, these processes result in greater net depolarization and therefore spiking for preferred motion than null motion. The number of SAC–SAC connections is markedly decreased in Pcdhg[1] retinas (*Figure 7I*). These connections persist in Pcdhg[0] retinas, but their efficacy is decreased because parallel SAC dendrites remain connected and inhibit each other, resulting in decreased inhibitory input from SACs to DSCGs for null motion and decreased antiparallel SAC–SAC inhibition (and thus increased SAC–DSCG inhibition) for preferred motion (*Figure 7F*). The autapses in Pcdhg[0] retina would act similarly to synapses between parallel dendrites, since autapsing dendrites are likely to point in similar directions (see Pcdhg[0] SAC image in *Figure 3*).

Second, inhibitory and excitatory currents in DSGCs are nearly simultaneous during null motion, allowing inhibition to veto excitation, whereas inhibition is delayed with respect to excitation during preferred motion, decreasing the power of the veto. A recent computational model argues that the delayed inhibition for preferred motion arises in part because anti-parallel SAC–SAC connections transiently suppress transmitter release from SACs to DSGCs (*Taylor and Smith, 2012*). Decreased inhibition, from loss of SAC–SAC connections in Pcdhg[1] retinas and decreased efficacy of SAC–SAC synapses Pcdhg[0] retinas, would thus be expected to decrease the delay, thereby blunting the response to preferred motion.

In summary, the spatial organization of SAC–SAC inhibition and SAC–DSGC inhibition work together to generate a direction-selective output from the retina. When self-avoidance, S/NSD, or synapse elimination is perturbed, SAC–SAC inhibition is rendered less effective and direction selectivity is degraded. Thus, our results demonstrate roles for these Pcdhg-dependent processes in computation of direction selectivity and provide new evidence in support of the hypothesis (*Lee and Zhou, 2006*; *Enciso et al., 2010*; *Taylor and Smith, 2012*; *Vaney et al., 2012*) that SAC–SAC connections play important roles in this computation.

## Materials and methods

### Animals

Animals were used in accordance with NIH guidelines and protocols approved by Institutional Animal Use and Care Committee at Harvard University. All mice were maintained on a C57BL/6 background. The lines used were reported previously: Pcdhg[fcon3] (*Lefebvre et al., 2008*; *Prasad et al., 2008*; *Lefebvre et al., 2012*), Chat[Cre] (*Rossi et al., 2011*), Thy1-stop-YFP line #15 (*Buffelli et al., 2003*), Mnx1::eGFP (here called HB9-GFP) (*Wichterle et al., 2002*; *Trenholm et al., 2011*), RC-stop-tdTomato (*Madisen et al., 2010*), and RC-stop-PcdhgC3-mCherry (*Lefebvre et al., 2012*). We generally used Chat[Cre] mice as homozygotes, because we found that this gave earlier and more even Cre activity at P1, when SAC dendrites are beginning to elaborate.

### Electrophysiology

Mice were dark adapted for at least 2 hr prior to euthanasia. Retinas were rapidly dissected under infrared illumination into room temperature, oxygenated (95% $O_2$, 5% $CO_2$) Ames medium and placed in a recording chamber on the stage of a custom built electrophysiology set up. Recordings were carried out in the same medium heated to 32–34°C. Fluorescent cells were identified with a brief (<40 ms) LED flash, overlaid onto infrared images, and targeted with electrodes. Recordings were made from SACs and vDSGCs using patch electrodes with resistance of 6–8 MΩ and 4–6 MΩ, respectively. For loose patch recordings, electrodes were filled with Ames medium. For intracellular recordings, electrodes were filled with intracellular solution containing the following (in mM): 120 Cs-Methanesulfonate, 10 Na-Acetate, 0.2 $CaCl_2$, 1 $MgCl_2$, 10 EGTA, 5 CsCl, 2 Mg-GTP, and 0.5 $Na_2$-GTP (pH 7.3). Intracellular recording solutions were supplemented with 5 mM QX314-Br for vDSGC voltage clamp recordings and 5 mM TEA-Br for SAC autapse recordings.

Paired connections were tested with 200 ms voltage steps from $V_h = -70$ mV to $+20$ mV in presynaptic SACs while postsynaptic SACs were held at $+30$ mV for all current size measurements and at potentials between $-70$ and $+30$ in 20 mV increments to establish I–V relationships. Approximately 10 voltage step repetitions were acquired for each pre-post pair and bidirectional measurements were made if recordings were sufficiently stable. Cells were analyzed in a semi-automated fashion and deemed connected using the following criteria: (1) Average traces had a peak in the first 30 ms after presynaptic stimulus onset that was >2 standard deviations from the baseline established in the 50 ms before stimulus onset, (2) current deflection was present in $\geq$ 80% of trials, (3) peak current had short latency (<12 ms) and fast rise time (10–90% rise time <4 ms). Each recording was checked after the fact for large baseline deviations or other anomalous signals.

Autapse recordings were evoked using a brief voltage step from $V_h = -70$ to $+60$ mV (2–4 ms) followed by a return to $-20$ mV. This stimulus activated some intrinsic currents in SAC that decayed in <100 ms. During this decay phase, a large fraction Pcdhg[0] SACs exhibited outward currents with synaptic latencies, rise times, and amplitudes that were blockable by the addition of 50 µM picrotoxin and thus autaptic currents. To analyze these recordings, we (1) fit the first 30 ms of each trace after returning to our holding potential of $-20$ mV with a double exponential curve, (2) looked for residuals of the fit >2 standard deviations of the pre-stimulus baseline in order to identify SACs that potentially made autapses, and (3) applied criteria used to find connected SAC pairs. We could not make reliable measurements of autapses in SACs from retinas younger than P21 due to large inward calcium currents evoked by depolarization. These currents were also apparent at even younger ages (P8) and may therefore be residually present from the ages at which SACs initiate and propagate retinal waves (see 'Discussion').

In loose patch spike recordings, action potentials were detected and analyzed using a simple thresholding criterion in MATLAB (Mathworks, Natick, MA). Spike histograms were made with 50 ms bins and used to find peak firing rates. DSIs and preferred directions of individual cells were calculated using the maximal firing rates elicited by moving visual stimuli in 8 directions ($\theta = 0°:45°:315°$) and vector sums were calculated as in *Kim et al. (2008)*.

## Visual stimuli

Light stimuli were presented using a modified DLP projector (Dell, Round Rock, TX) suspended underneath the microscope stage with a custom substage lens system focused onto the retinal photoreceptors. Monochrome light was used (wavelength peak = 405 nm) at a background intensity $5 \times 10^2$ R*/rod/s set using neutral density filters. Visual stimuli were presented at 100:1 positive contrast and patterns generated using Psychophysics Toolbox in MATLAB and are available as *Source code 1*. All stimuli were centered on the cell body of recorded neurons. Spot flash stimuli were 300 µm-diameter circles. Moving bars were 1000–1500 µm long and 300 µm wide, traveled at 1000 µm/s, presented moving along their long axis in 8 directions, and rotated by 135° with each presentation. At the speeds we used for our visual stimuli, nonlinear dendritic processes contributing to directional tuning are not observed in HB9-GFP vDSGCs (*Trenholm et al., 2011*). A minimum of 4 repetitions were presented for each stimulus.

## Data acquisition and analysis

Electrophysiological recordings were acquired using a Multiclamp 700B Amplifier (Axon Instruments, Molecular Devices, Sunnyvale, CA) at 20 kHz. Acquisition was controlled by custom LabView software (National Instruments, Austin, TX) and is available as *Source code 2*. Data were analyzed using custom written MATLAB software available as *Source code 3* and displayed in IgorPro (Wavemetrics, Portland, OR). All statistics were calculated in MATLAB. Pairwise comparisons were made using two-tailed t-test, and multiple samples were compared using one-way analysis of variance. Errors on connection probability were calculated using the variance of the binomial distribution. The specificity of reciprocal connections between neighboring SACs was assessed by comparison with Monte Carlo simulations using recorded connection probabilities.

Latencies for paired recordings from SACs and directional voltage-clamp recordings from vDSGCs were measured by fitting the rising phase of each current using a Boltzmann function in IgorPro and finding the intersection this line with the baseline. Latencies of autaptic currents (after automated

detection) were calculated manually and taken from the beginning of the short voltage steps to + 60 mV.

## SAC and DSGC fills and histology

SACs and DSGCs were filled through patch electrodes using methods described above. Alexa Fluor 488 hydrazide (200 μM) was added to the intracellular recording solution, and recordings were maintained for ~20 min in current-clamp mode while maintaining a negative holding potential (<−60 mV). After individual cells were filled, retinas were either imaged live (to measure SAC dendritic radius) or immediately placed in fixative and processed for histology.

Mice used exclusively for histology were euthanized by intraperitoneal injection of pentobarbital or euthasol and either enucleated immediately or transcardially perfused with Ringer's solution followed by 4% paraformaldehyde (PFA) in PBS. Eye cups were removed and fixed in 4% PFA in PBS on ice for 1 hr then rinsed with PBS. Retinas were analyzed as whole mounts or cryosections as described previously (*Lefebvre et al., 2012*). Whole mount retinas were incubated in blocking buffer (0.5% Triton-X-100, 5% normal donkey serum in PBS) for 1–2 hr at room temperature, then incubated for 5–7 days at 4°C with primary antibodies. For cryosections, fixed retinas were incubated with 30% sucrose/PBS for >2 hr (until they lost buoyancy), frozen, and sectioned at 20 μm in a cryostat. Sections were blocked with 5% donkey serum/0.5% Triton X-100/PBS for 1–2 hr at room temperature, with primary antibodies overnight at 4°C, and with secondary antibodies for 2 hr at room temperature. Whole mount retinas or sections were mounted onto glass slides using Fluoromount G (Southern Biotech). The following primary antibodies were used: chick anti-GFP (1:500, Abcam); rabbit anti-DsRed (1:1000, Clontech); goat anti-choline acetyltransferase (ChAT) (1:400, Millipore); goat anti-VAChT (1:1000, Promega); rabbit anti-Calbindin (1:2500, Swant); rabbit anti-CART (1:1000, Phoenix); mouse anti-Brn3a (1:1000, Millipore); goat anti-Chx10 (1:200, Santa Cruz); and mouse anti-AP2 (1:1000, DSHB). Nuclei were labeled with TO-PRO3 (1:3000, Invitrogen). Secondary antibodies were conjugated to Alexa Fluor 488, Alexa Fluor 568 (Invitrogen), or DyLight 649 (Jackson ImmunoResearch) and used at 1:1000.

Immunofluorescence samples were imaged using Olympus FV1000 confocal microscope using 488, 568, and 647 lasers with a z-step size of 1.0 μm. FIJI (NIH) was used to analyze confocal stacks and generate maximum intensity projections. ON and OFF dendrites of DSGCs were separated using depths in the inner plexiform layer and corresponding SAC bands. Retinal orientations were maintained throughout.

## Acknowledgements

We thank members of the Sanes lab (official and adopted) for useful discussions and comments on this manuscript, Julie Lefebvre for guidance during initial stages of the project and comments on this manuscript, Ed Soucy and Joel Greenwood for assistance in construction of electrophysiology set-up, Micheal Do for advice on physiological methods, and Tim Dunn for assistance with data analysis software. Funding was provided by NIH grants T32 EY007110 and F31 NS078893 (DK) and RO1 EY022073 (JRS).

## Additional information

### Funding

| Funder | Grant reference | Author |
| --- | --- | --- |
| National Institutes of Health (NIH) | R01 EY022073 | Joshua R Sanes |
| National Institutes of Health (NIH) | T32 EY007110 | Dimitar Kostadinov |
| National Institutes of Health (NIH) | F31 NS078893 | Dimitar Kostadinov |

The funder had no role in study design, data collection and interpretation, or the decision to submit the work for publication.

### Author contributions

DK, Conception and design, Acquisition of data, Analysis and interpretation of data, Drafting or revising the article; JRS, Conception and design, Analysis and interpretation of data, Drafting or revising the article

## Ethics

Animal experimentation: Animals were used in accordance with NIH guidelines and protocols approved by Institutional Animal Use and Care Committee at Harvard University (IACUC protocol #24-10). Animals were euthanized by intraperitoneal injection of pentobarbital or euthasol.

## Additional files

### Supplementary files

- Source code 1. Visual stimuli.

- Source code 2. Microscope acquisition.

- Source code 3. Electrophysiology analysis.

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
