## [Decision Letter]

Thank you for submitting your work entitled “Protocadherin-dependent dendritic self-avoidance regulates neural connectivity and circuit function” for peer review at *eLife*. Your submission has been favorably evaluated by Eve Marder (Senior editor), a Reviewing editor, and three reviewers.

The reviewers have discussed the reviews with one another and the Reviewing editor has drafted this decision to help you prepare a revised submission.

The following individuals responsible for the peer review of your submission have agreed to reveal their identity: Kang Shen (Reviewing editor) and Alex Kolodkin (peer reviewer). A further reviewer remains anonymous.

All three reviewers expressed high level of enthusiasm towards this manuscript and found this manuscript very interesting and important. Reviewer 1 wrote: “This is an excellent paper, certainly of the quality one sees in Nature, Cell or Science.” Another reviewer wrote: “The demonstration of the need for self-avoidance and self/non-self-discrimination for optimal circuit function is a very nice in situ demonstration of the importance of these processes during neural development.”

One of the reviewers raised the point that it might be possible to mine the single cell expression data to further test the notion that each SAC expressing a few of the gamma protocadherin isoforms, which are stochastic, or at least minimally overlapping amongst neighboring cells. If you have data that speak to this issue already, it might be helpful to include this data. If it takes significant time and effort to get this data, the other reviewers and I agree that this would be outside of the scope of this study.

Reviewer #3 stressed that the only major criticism of this paper is that it is premised on each SAC expressing a few of the gamma protocadherin isoforms, which are stochastic, or at least minimally overlapping amongst neighboring cells. This is a very attractive idea, but it remains untested. Dr. Sanes was recently an author on a paper using Drop-seq to profile single cell transcriptomes from retinal neurons, including 289 SACs, identified by their Chat expression (Macosko et al., Cell, 2015). This data should be mined to determine if these cells do indeed express only a few Pcdhg isoforms, and if they are indeed random. If, for example, it was found that all SACs express all Pcdhg exons, or if there were two distinct clusters perhaps representing ON versus OFF SACs, it would dramatically change the model underlying the results presented here. While the positions of these neurons in the retina are not available, the results should at least be consistent with the proposed mechanism. Quantification of expression level from single cell data may also be problematic, but again, one would hope that the general results are at least consistent with model proposed.

---

## [Author Response]

*Reviewer #3 stressed that the only major criticism of this paper is that it is premised on each SAC expressing a few of the gamma protocadherin isoforms, which are stochastic, or at least minimally overlapping amongst neighboring cells. This is a very attractive idea, but it remains untested. Dr. Sanes was recently an author on a paper using Drop-seq to profile single cell transcriptomes from retinal neurons, including 289 SACs, identified by their Chat expression (Macosko et al., Cell, 2015). This data should be mined to determine if these cells do indeed express only a few Pcdhg isoforms, and if they are indeed random. If, for example, it was found that all SACs express all Pcdhg exons, or if there were two distinct clusters perhaps representing ON versus OFF SACs, it would dramatically change the model underlying the results presented here. While the positions of these neurons in the retina are not available, the results should at least be consistent with the proposed mechanism. Quantification of expression level from single cell data may also be problematic, but again, one would hope that the general results are at least consistent with model proposed*.

The idea of sparse, stochastic expression of Pcdhgs is based on convincing data of Yagi et al. from Purkinje cells. We agree that it remains unproven for SACs, but that is not for lack of effort. First, as part of his work on the preceding paper (41), Dimitar applied to SACs the “split single cell RT-PCR” protocols that Yagi and colleagues had devised to analyze Pcdhg expression in Purkinje cells (Esumi, Nature Genetics, 2005; Kaneko et al., JBC, 2006). He detected Pcdhg expression in groups of ∼10 cells but was unable to obtain consistent results from single cells. Although we cannot rule out ineptitude as a factor, we believe a major problem is that the volume of SAC is ∼3% that of Purkinje cells, so copy number is correspondingly decreased. Second, in experiments done for another purpose, we recently profiled 96 isolated SACs using a MALBAC protocol (Chapman, PLoS One, 2015). We detected 18 of the 22 Pcdhgs. Each was expressed by a subset of the 96 cells, ranging from 3 to 26 with an average of around 15. This is consistent with the combinatorial expression that Yagi showed in Purkinje cells and that we propose for SACs. At face value, we could conclude that around 3 isoforms were expressed per cell (15x22/96). However, we do not feel comfortable reporting these data in a formal way because we have no way of knowing whether apparently negative cells represent true negatives or insufficient capture efficiency or sequencing depth. The beauty of Yagi's method is that the “split cell” strategy provided assurance on this point – but as I already said, we were unable to get this method to work on SACs. Finally, as the reviewer points out, we recently used Drop-Seq to obtain transcriptomic data from around 33,000 unsorted retinal cells, of which 289 were identifiable as SACs (Macosko, Cell, 2015). Here Pcdhg detection was hampered by the low sequence depth and the fact that reads were heavily biased to the 3' end, which is shared by all of the Pcdhgs. (That is, only reads from the 5' half of the transcript are diagnostic.) In fact, we detected almost no Pcdhgs in SACs. Pcdhbs (which have unique 3’ ends) were detected, consistent with the 3’ bias. In short, we have worked hard to test the idea of sparse, combinatorial expression of Pcdhgs in SACs and although the data we have obtained are consistent with this model, we are unable to make any definitive statement at this time.